# L-DOPA and Droxidopa: From Force Field Development to Molecular Docking into Human β_2_-Adrenergic Receptor

**DOI:** 10.3390/life12091393

**Published:** 2022-09-06

**Authors:** Andrea Catte, Akash Deep Biswas, Giordano Mancini, Vincenzo Barone

**Affiliations:** Scuola Normale Superiore, Piazza dei Cavalieri 7, 56126 Pisa, Italy

**Keywords:** catecholamines, drugs, adrenergic receptors, L-DOPA, Droxidopa, force field parameterization, quantum mechanical calculations, molecular docking, molecular dynamics simulations

## Abstract

The increasing interest in the molecular mechanism of the binding of different agonists and antagonists to β_2_-adrenergic receptor (β_2_AR) inactive and active states has led us to investigate protein–ligand interactions using molecular docking calculations. To perform this study, the 3.2 Å X-ray crystal structure of the active conformation of human β_2_AR in the complex with the endogenous agonist adrenaline has been used as a template for investigating the binding of two exogenous catecholamines to this adrenergic receptor. Here, we show the derivation of L-DOPA and Droxidopa OPLS all atom (AA) force field (FF) parameters via quantum mechanical (QM) calculations, molecular dynamics (MD) simulations in aqueous solutions of the two catecholamines and the molecular docking of both ligands into rigid and flexible β_2_AR models. We observe that both ligands share with adrenaline similar experimentally observed binding anchor sites, which are constituted by Asp113/Asn312 and Ser203/Ser204/Ser207 side chains. Moreover, both L-DOPA and Droxidopa molecules exhibit binding affinities comparable to that predicted for adrenaline, which is in good agreement with previous experimental and computational results. L-DOPA and Droxidopa OPLS AA FFs have also been tested by performing MD simulations of these ligands docked into β_2_AR proteins embedded in lipid membranes. Both hydrogen bonds and hydrophobic interaction networks observed over the 1 μs MD simulation are comparable with those derived from molecular docking calculations and MD simulations performed with the CHARMM FF.

## 1. Introduction

The Guanine Nucleotide-Binding Protein Coupled Receptors (GPCRs) form the largest family of human trans-membrane proteins and take part in a wide range of critical biological functions including sight, sensation and neurological transmission [1,2,3]. There are several activities of the GPCRs in the body, including cognitive responses [4], cardiovascular functions [5] and the growth and development of cancer [6]. Thirty-five percent of all commercially available drugs in the United States and throughout the world target GPCRs because of their relevance in many human disorders [7,8,9]. In all GPCRs there are seven transmembrane α helices (TM-I-TM- VII), which are linked by extracellular (ECL1-ECL3) and intracellular (ICL1-ICL3) loops [10,11]. In addition to hormones and therapeutics, a large variety of other extracellular molecules can act as agonists or antagonist on GPCRs, with the former often causing conformational changes in proteins associated with specific activities [12]. Furthermore, GPCRs are being viewed as the allosteric machinery that can be triggered by ions, lipids, cholesterol and water [13,14]. Moreover, due to technological breakthroughs in crystallization technologies during the last two decades, new X-ray crystal structures of GPCRs are being reported with an exponential rate [3,15]. In addition, over 150 GPCR X-ray structures published in the www.rcsb.org Protein Data Bank (accessed on 5 January 2020) are in complex with ligands [16]. At the same time, homology models of GPCRs have provided a molecular representation of more than 10% of the GPCR super-family. Structure-based drug discovery (SBDD) relies on understanding receptor-drug interactions at the atomic level, so molecular docking and molecular dynamics simulations have become widespread tools for drug design, measuring binding affinity, revealing reaction mechanisms and protein-ligand interactions, in addition to understand the GPCR structure and dynamics [17,18,19,20,21,22].

To activate the class A GPCR β_2_-adrenergic receptor (ADRB2 or β_2_AR), which is found in pulmonary and cardiac myocyte tissues, hormones, such as adrenaline and noradrenaline together with the neurotransmitter dopamine, are required. There has been some progress in understanding the inactive state of adrenergic receptors thanks to the identification of the first high resolution X-ray crystal structures of β_2_AR bound to inverse agonist (-)-carazozol (PDB ID: 2RH1) [23,24] and antagonist (-)-timolol (PDB ID: 3D4S) [25]. In addition, β_2_AR stabilized with a nanobody and a nucleotide-free Gs protein heterotrimer has its first agonist-bound active-state X-ray crystal structure determined [26,27]. Moreover, the β_2_AR X-ray crystal structure, released in 2013 (PDB ID: 4LDO) [28], has served as a structural template for the investigation of the binding conformations and affinity of several endogenous agonists and antagonists. Studies using fluorescence spectroscopy have shown that catecholamines, including adrenaline, noradrenaline and dopamine may modify the β_2_AR’s conformation, leading to the formation of distinct adrenergic receptor intermediate states [29,30,31,32]. There have been more recent confirmations of the β_2_AR’s structural heterogeneity by NMR spectroscopy, which reveals that it may exist in three different states: active, intermediate and inactive [33]. Human β_2_AR structures in association with 1365 ligands, 75 of which are drugs, may be found in the biggest database of GPCR structures and mutations (www.gpcrdb.org (accessed on 20 December 2019)) [7]. Among the drugs that interact with β_2_AR is droxidopa (L-DOPS), which is an L-serine with a 3,4-dihydroxyphenyl group replaced at the beta position [34]. The amino acid levodopa (L-DOPA) is structurally identical to L-DOPS and a cathecolamine L-tyrosine derivative, which is an intermediate product of the biosynthesis of dopamine [35]. Since the symptoms of Parkinson’s disease (PD) are related to a progressive reduction of dopamine levels in the brain, dopamine and drugs able to safely cross the blood–brain barrier have been used to increase the dopaminergic function in patients with PD. Due to the inability of dopamine to cross the blood–brain barrier, L-DOPA in combination with carbidopa, which is an inhibitor of extracerebral dopa decarboxylase (IEDD), has also been employed in the treatment of PD [35]. Recently, PD and neurogenic orthostatic hypotension have been both treated with droxidopa, a noradrenaline precursor, which has been authorized in Japan and is now being tested in Europe, the United States, Australia and Canada. More recently, prodrugs of L-DOPA and carbidopa, namely levodopa-4${}^{\prime }$-phosphate (foslevodopa) and carbidopa-4’-phosphate (foscarbidopa), have been employed in successful clinical trials on patients with advanced PD in the United States [36,37] and are going to be tested in the United States and Australia [38].

Here, we report the results of the force field (FF) development of L-DOPA and Droxidopa molecules, and their MD simulations in water. All FF parameters were obtained through high-level QM calculations, according to the procedure implemented in the JOYCE program [39]. All MD simulations were performed using an FF parameterized ad hoc for each solute of interest to increase the accuracy of the conformational sampling. In addition, we show the results of the molecular dockings of these exogenous catecholamines into the human β_2_AR X-ray crystal structure released by Ring et al., in 2013 (PDB ID: 4LDO) and their comparison with observations from the full agonist adrenaline. Each wholly flexible ligand was docked into the β_2_AR binding pocket following two different approaches: (1) a rigid receptor model derived from the PDB coordinates of the β_2_AR X-ray crystal structure (PDB ID: 4LDO), and (2) a β_2_AR receptor with flexible side chains of key amino acids in its binding site. Both approaches show and confirm that the pocket conformation is compatible with the binding of adrenaline, suggesting that a similar binding mode can be predicted for L-DOPA and Droxidopa. The characteristic network of hydrogen bonds of catecholamines is also preserved, showing the equal importance of β_2_AR amino acids interacting with head (Ser203/Ser204/Ser207) and tail (Asp113/Asn312) moieties of every analyzed ligand. However, other β_2_AR residues, such as Thr118 and Tyr316, also play a remarkable role in the binding of L-DOPA and Droxidopa. Moreover, the binding affinities calculated from the molecular docking of adrenaline are in good agreement with experimental values, allowing a more reliable comparison of this property estimated for L-DOPA and Droxidopa non-natural ligands. The findings of this article are presented in the following sections starting from the development and validation of a new FF for L-DOPA and Droxidopa, and then proceeding to a discussion of the molecular docking results.

## 2. Materials and Methods

### 2.1. Force Field Development Protocol

The FF parameter set of adrenaline, L-DOPA and Droxidopa ligands were obtained by fitting energies, gradients and Hessian to the results of Electronic Structure calculations, as described in the Joyce/Ulysses procedure [39,40]. Electronic structure calculations were carried out with the Gaussian16 suite of programs [41,42], using Density-Functional Theory (DFT) [43] with the hybrid Becke3LYP functional (B3LYP) [44] in conjunction with the jul–cc-pVDZ basis set [45], taking into account solvent effects by means of the Conductor-like Polarizable Continuum Model (C-PCM) [46,47] and setting water as a reference solvent (Appendix A). The CM5 method based on Hirschfeld partitioning [48] was used to determine atomic charges in view of its near invariance for different quantum chemical models, remarkable reproduction of molecular dipole moments consistence with the latest revisions of the OPLS all atom (AA) force field [49,50,51,52] used for dispersion–repulsion interactions (Appendix A). Dihedral angles were assigned using GaussView version 6 [53]. Fitting of QM torsional profiles by linear combinations of cosine functions (Figure 1), were performed with Grace version 5.1.25 (Paul J. Turner, Portland, OR, USA). Dihedral angles γ and θ of adrenaline, γ of L-DOPA and η of Droxidopa were refined manually after the initial estimate.

Molecular dynamics (MD) simulations of protonated adrenaline, zwitterionic L-DOPA and Droxidopa in water were performed using Gromacs [54,55,56]. The SPC model [57] was used to describe water molecule (adrenaline or L-DOPA or Droxidopa) interactions. All the systems were simulated in the NVT ensemble with periodic boundary conditions, with an integration time step of 2 fs. The temperature was kept constant at 300 K with velocity-rescaling temperature coupling [58]. The LINCS algorithm was used to constrain all bonds [59]. The Particle Mesh Ewald (PME) method was used to compute long-range electrostatic interactions with grid search and a cut-off radius of 1.1 nm [60]. For each molecule, 20 ns NVT MD simulations in vacuum were performed using JOYCE-derived topology. Then, MD simulation production runs of adrenaline, L-DOPA and Droxidopa in SPC water were performed for 100 ns at 300 K. The analysis of MD simulations was performed with Gromacs analytical tools (gmx angle, gmx distance etc.). Then, 20 ns structures of each ligand simulated in water were used as starting conformations for the molecular docking into rigid and flexible β_2_AR models.

### 2.2. β_2_AR Molecular Modeling

The atomistic model of the TMD of human β_2_AR was built up from the X-ray crystal structure of the protein active state conformation (PDB ID: 4LDO) [28]. Amino acid residues 1–28 and 343–413, which are not present in the X-ray structure, were not included following an approach similar to the one employed by Rosenbaum et al., 2011 [61]. The ICL3 domain intracellular residues 232–262, which were not employed by Rosenbaum et al., in 2011 and modeled by Dror et al., 2009 [61,62], were taken from a β_2_AR homology model built up using 2RH1 as a template and reported in the Sali Laboratory ModBase database of comparative protein structure (Figure 2) [63]. 2RH1 is an excellent template structure for β_2_AR due to its high sequence identity (90%) and a good RMSD of C_α_s of residues 29–231 and 263–342 (2.5 Å) with respect to 4LDO. T4 lysozyme (T4L) and Nanobody 6B9 (Nb6B9) amino acids were removed, and N- and C- termini were made neutral using acetyl and methylamino groups, as reported in previous studies [61,62,64,65]. Water molecules and adrenaline ligands were not included in the AA model. The sequence of human β_2_AR (https://www.uniprot.org/uniprot/P07550 (accessed on 20 December 2019)) was also reproduced by mutating the four engineered mutations (Met96Thr, Met98Thr, Asn187Glu and Cys265Ala) present in the receptor’s X-ray crystal structure released by Ring et al., in 2013 to their native amino acid forms [28].

All lysines and arginines were protonated, while all aspartates, glutamates and histidines (HSD) were deprotonated except for Glu122 (GLH), Asp130 (ASH) and His172 (HSP) residues, as previously reported by Dror et al., in 2009 and 2011 [62,64]. Since the β_2_AR all atom structure is in its active conformation, Asp79 was deprotonated because the protonation state has been suggested to be stable upon activation [66,67]. The cysteine residues located in ECL2 (Cys184- Cys190) and TM-III (Cys106-Cys191) domains of the β_2_AR model were modified to form disulfide bonds by deleting hydrogens bound to SG atoms [68]. The building up of the final β_2_AR all atom structure was performed using VMD 1.9.4 and its psfgen plugin [69].

### 2.3. Molecular Docking Protocol

Initially, we performed an alignment of the geometry optimized structures of L-DOPA (DAH) and Droxidopa (DRO) molecules to adrenaline (ALE) in the β_2_AR X-ray crystal structure (PDB ID: 4LDO), modified as described above, to produce an improved docking grid box around the receptor’s binding pocket. The β_2_AR model shown in Figure 2 was used as a rigid receptor. In the flexible model of β_2_AR, protein residues of the binding pocket (Asp113, Val117, Thr118, Phe193, Thr195, Ser203, Ser204, Ser207, Asn293, His296, Asn301, Tyr308, Asn312 and Tyr316: see ref. [28]) forming polar and apolar interactions with adrenaline were considered flexible, using a similar approach to that reported by Tosso et al., 2020 in the molecular docking of dopamine to the D_2_DR receptor [70]. In the case of L-DOPA, His296 and Asn301 were excluded from the list of flexible residues because the total number of torsional degrees of freedom, including those of the ligand (6), would have exceeded the maximum number allowed by AutoDock Tools 1.5.6, the graphical user interface of AutoDock 4.2.6 (AD4) for generating consistent docking results [71]. For Droxidopa His296, Asn301 and Val117 were not included in the list of flexible amino acids because of the additional torsional degree of freedom of this ligand as compared to L-DOPA. In both receptor models, the X-, Y-, and Z-axes dimensions of the docking box grid were 50, 50 and 50, respectively, and a resolution of 0.375 Å was employed in the active site region. The grid box size was 18.75 Å, which is more than 2.9 times the radius of gyration of both L-DOPA (2.77 Å) and Droxidopa (2.91 Å) molecules, in good agreement with recommendations by Feinstein et al., in 2015 [72]. Non-polar hydrogen atoms were merged into heavy atoms and Gasteiger charges were assigned to each molecule. During each docking run, all torsion angles of flexible amino acids and ligands were free to rotate. Four-hundred poses were generated using the maximum number of generations and energy evaluation of 27,000 and 5e7, respectively, for both rigid and flexible β_2_AR models, as previously reported for the molecular docking of dopamine to the D_2_DR receptor [70]. The Virtual Screening analysis of the final docked conformations was performed with the AD4 pythonsh command using a tolerance of 2 Å root mean square deviation (RMSD) for the clustering [73,74].

The binding free energy ($\Delta G_{\mathrm{bind}}=-{\mathrm{RTlnK}}_{i}$) (BFE) and inhibition constants (K_i_) of lowest energy structures, estimated by AD4 with the Virtual Screening method, were employed to calculate the binding affinities of each ligand [73]. Since experimental binding affinities of adrenaline and other endogenous catecholamines were measured at 310.15 K, AD4 BFEs estimated at 298.15 K were also corrected, resulting in an increase of 0.2 units for the corresponding binding affinity (pK_d_) of each ligand at the physiological temperature [75].

Two sets of ligand heavy atom RMSDs were calculated using the X-ray crystal structure of adrenaline as a reference: one employed the original coordinates of the ligand and the other, defined as superimposed RMSD, involved a structural alignment of the ligand with the X-ray structure of adrenaline. Superimposed RMSDs were also calculated using the lowest energy conformations derived from rigid and flexible molecular dockings for each ligand.

To validate the molecular docking protocol, adrenaline was redocked into the β_2_AR X-ray crystal structure (PDB ID: 4LDO). Two sets of redocking experiments were performed using β_2_AR rigid and flexible models. From the rigid docking set, the lowest energy structures of each cluster were selected to perform ten independent runs. The consistency of the docking results was considered achieved when at least 70% of the individual runs generated conformations clustered within an RMSD of < 0.5 Å relative to the overall best energy pose of the ligand. For both protonation states of adrenaline, rigid docking runs using the same coordinates of the receptor X-ray crystal structure produced conformations (heavy atoms superimposed RMSD of 0.2 Å) and BFEs (−6.4 kcal/mol and −8.1 kcal/mol for neutral and protonated adrenaline, respectively) similar to those of the lowest energy conformations from ten independent runs [76]. The BFE of adrenaline in the X-ray crystal structure was estimated with the epdb option of AD4.

In order to improve the performance of AD4 runs, the multilevel parallel version of AutoDock 4.2 was also employed to perform multiple independent runs of various ligands [77]. RMSDs of ligand heavy atoms were calculated using AutoDock Tools 1.5.7 [78].

The number of hydrogen bonds and hydrophobic interactions between a ligand and the receptor were measured using the protein–ligand interaction profiler (PLIP) [79]. Images were prepared using VMD 1.9.4, AutoDock Tools 1.5.6 and LigPlot+ version 2.2 [80].

### 2.4. MD Simulations of β_2_AR-Catecholamine Complexes

In order to test the quality of FFs developed for the different ligands, we also performed 1 μs AA MD simulations of β_2_AR-catecholamine complexes embedded in solvated lipid bilayers composed of 260 palmitoyloleoylphosphatidylcholine (POPC) and 65 cholesterol (CHOL) molecules (20% CHOL) using the OPLS AA FFs for proteins [52] and lipids [81,82,83], following an approach similar to that reported by Biswas et al., in 2021 [76].

The contact maps of β_2_AR residues with each ligand were generated with the Timeline plugin version 2.3 of VMD 1.9.4 [69]. A ligand-protein contact was counted only when at least one atom of each catecholamine was within 5 Å of any atom of the receptor.

BFEs of ligands were extracted from MD simulations of β_2_AR-catecholamine complexes using Molecular Mechanics Poisson–Boltzmann Surface Area (MM-PBSA) calculations [84].

## 3. Results

### 3.1. Force Field Parameterization and Md Simulations of L-Dopa and Droxidopa

Following the protocol described in the methods section, the QM torsional profiles of each dihedral angle of the zwitterionic L-DOPA were accurately reproduced by the corresponding JOYCE FF functions (Figure 3). It is worth noting that both QM and JOYCE torsional profiles of α, β, γ and δ dihedrals were very similar to those of zwitterionic Tyrosine reported by Del Galdo et al., in 2018 [85]. Concerning the torsions of the charged groups (α and β dihedrals) in the zwitterionic condition, the agreement between QM and JOYCE torsional profiles was better than the already satisfactory one reported by Del Galdo et al., in 2018 for the closely related L-Tyrosine amino acid, indicating the good quality of the FF parameterization of L-DOPA [85].

Similar to L-DOPA, the QM torsional profiles of the dihedrals of zwitterionic Droxidopa were in good agreement with JOYCE FF functions (Figure 4). In particular, QM and JOYCE torsional profiles of α, β, γ, δ, ϵ and η dihedrals of Droxidopa were very similar to those observed for L-DOPA (Figure 3).

Moreover, we also generated OPLA AA FF parameters for adrenaline, whose QM torsional profiles were accurately reproduced by the corresponding JOYCE FF parameters for all dihedral angles. QM and JOYCE torsional profiles of α, γ, δ, ϵ and η dihedrals of adrenaline were very similar to those observed for L-DOPA and Droxidopa ligands (Appendix A). A complete list of the most representative bond lengths, bond angles and dihedral angles of each ligand is provided in the SI of this work (Appendix A). To test the FFs of adrenaline, L-DOPA and Droxidopa, we performed MD simulations of each parameterized ligand both in vacuum and in aqueous solution. It is worth noting that all the dihedral distributions are comparable in the gas phase and in aqueous solution for both L-DOPA and Droxidopa molecules (Figure 5 and Figure 6). Similar results were also observed for adrenaline (Appendix A).

Additionally, we also observed a good agreement between the fittings of QM torsional energy profiles subjected to the subtraction of electrostatic and Lennard–Jones contributions (Appendix A) and correspondent QM and JOYCE for each dihedral of adrenaline (Appendix A), L-DOPA (Appendix A) and Droxidopa (Appendix A). The profiles of the Helmoltz free energy variation as a function of each of the dihedral angles were also similar to the corresponding QM energy for both L-DOPA and Droxidopa (Figure 7 and Figure 8). Similar behavior of Helmholtz free energy variations was also found for each dihedral of adrenaline (Appendix A). The Helmholtz free energy variation (Δ*A*) was computed from the MD ensemble probability distribution of the generic θ dihedral coordinate, according to the equation:$$\Delta A\left(\theta \right)=-kTln(\rho \left(\theta \right)/\rho_{0})$$
where *k* is the Boltzmann constant, *T* is the temperature, ρ(θ) is the dihedral probability distribution having ρ_0_ as a maximum value (missing Δ*A* points in the figures imply that no sampling could be found for the corresponding geometry).

Each profile obtained using as a statistical ensemble the MD simulation performed employing Joyce-derived FF parameters is also compared with the one obtained with the use of the OPLS AA FF (see Methods). The differences between QM energy and MD Δ*A* profiles can be ascribed to the entropic contribution, which is only present in the free energy, as previously observed by Del Galdo et al., in 2018 for zwitterionic Tyrosine [85]. Despite these discrepancies, this comparison can be used to analyze, at least qualitatively, the accuracy of the FF for the different dihedral angles.

### 3.2. Validation of the β_2_AR Model: Redocking of Adrenaline

In our recent work, we verified the ability of the β_2_AR X-ray crystal structure (PDB ID: 4LDO) to generate biologically relevant binding conformations of adrenaline by redocking the endogenous ligand into rigid and flexible receptor models, employing a similar approach to the one reported by Katritch et al., in 2009 [76,86]. At the physiological pH of 7.4, adrenaline is mostly present in its protonated form (94.7%); however, due to an experimental pKa of 8.52, the percentage of neutral forms is not negligible (5.3%) [87]. The effect of the protonation state of this endogenous ligand on its interaction with β_2_AR has been recently investigated by performing the redocking using both species [76]. We have shown that the neutral form of adrenaline is about 2 kcal/mol less stable than its more biologically relevant protonated form. This energy difference arises from the higher number of hydrogen bonds formed by the protonated species with residues in the β_2_AR binding pocket as compared to the neutral form of the ligand [76]. Hereafter, adrenaline refers to the protonated form of the endogenous catecholamine. In particular, the redocking of adrenaline into rigid and flexible β_2_AR models generates conformations different from that of the ligand in the X-ray crystal structure, exhibiting ligand heavy atoms RMSDs of 2.2 Å and 2.4 Å, respectively (Figure 9A,B).

Although ligand–receptor hydrogen bonds of the ethanolamine tail of the redocked adrenaline with the anchor site formed by amino acids Asp113/Asn312 are similar to those observed in the X-ray crystal structure, the catechol head of the molecule interacts with different residues, namely Ser203/Asn293 and Thr118/Ser207 in crystal and redocked structures, respectively (Figure 9A,B). These results mainly arise because there is a substantial energetic difference between the free energies of binding of redocked and crystal structure ligands. According to experiments, the BFE of adrenaline ranges from −8.3 kcal/mol to −8.9 kcal/mol, as calculated from binding affinities of adrenaline reported by Del Carmine et al., in 2004 and 2002, indicating that the structure of adrenaline from the rigid redocking is almost in its more energetically stable conformation [75,88]. Interestingly, rigid and flexible docking lowest energy conformations of adrenaline are also structurally similar, as shown by the ligand heavy atom RMSD of 0.6 Å (superimposed RMSD = 0.3 Å). However, the average BFE of adrenaline for the lowest energy structure from the Virtual Screening of the flexible docking is −12.5 kcal/mol, indicating that this conformation of the ligand is not only much more stable than the one obtained from the rigid docking but it also has a pK_d_ better than the experimental values reported in the literature (Table 1) [75,88].

### 3.3. L-DOPA Binding to the β_2_AR Receptor

The docking of L-DOPA to a rigid β_2_AR model generated 70% of the largest cluster conformations from ten independent runs showing an adrenaline-like interaction of catechol and ethanolamine moieties of the ligand with the binding pocket anchor sites Ser203/Ser204/Ser207 and Asp113/Asn312, respectively (Figure 10A and Table 2). These conformations of the catecholamine displayed a pK_d_ of 4.7 and an average BFE of −6.4 kcal/mol, which were similar to the values we recently reported for neutral adrenaline [76]. Interestingly, we also observed that the docking of L-DOPA to a rigid model of the receptor also yielded the lowest energy conformations unable to form hydrogen bonds in an adrenaline-like fashion [76].

The molecular docking of L-DOPA into a flexible β_2_AR model yielded more ligand–receptor hydrogen bonds than those observed in the lowest energy conformation of the catecholamine produced by the rigid docking (Figure 10B and Table 2). In particular, the catechol head of the ligand also formed an additional hydrogen bond with the binding site amino acid Thr118. The lowest energy conformations showed an average BFE of −12.8 kcal/mol, corresponding to a pK_d_ of 9.4 and being comparable to that observed for the endogenous ligand adrenaline docked into a flexible β_2_AR model (Table 1).

We have recently observed that Autodock Vina (Vina) [89] generated best conformations of L-DOPA docked into rigid and flexible β_2_AR models displaying average BFEs of −7.3 and −7.9 kcal/mol, respectively, which were different from correspondent AD4 results (−6.4 and −12.8 kcal/mol, respectively) (Appendix A) [76]. In the flexible β_2_AR model, this difference in the stability of the ligand is due to the higher number of hydrogen bonds formed by the best AD4 conformation as compared to the best Vina structure (Appendix A). However, the Vina conformation, as observed in the docking to a rigid β_2_AR model, was still structurally closer to adrenaline’s conformation in the X-ray crystal structure than the best AD4 pose (Appendix A).

### 3.4. Droxidopa Binding to the β_2_AR Receptor

The rigid docking of Droxidopa to β_2_AR generated the lowest energy binding poses comparable to those observed for adrenaline, in which the ethanolamine moiety of the ligand was bound to the Asp113/Asn312 anchor site and its catechol moiety formed bonds only with Ser207 in the TM-V domain of the receptor (Figure 11A). It is worth noting that the lowest energy pose showed an average BFE of −7.4 kcal/mol (best conformations resulting from ten independent docking runs), which is very similar to the one we recently observed for noradrenaline (−7.6 kcal/mol), of which Droxidopa is a precursor, confirming the validity of the molecular docking protocol [76].

Moreover, the pK_d_ of Droxidopa (5.4) is also in good agreement with experimental and computational values available for noradrenaline [75,76,86,88]. Similarly to L-DOPA, the best binding poses of Droxidopa docked in the flexible β_2_AR model formed more hydrogen bonds with both anchor sites of the binding pocket of the receptor as compared to the ligand bound to the rigid model (Figure 11B). The lowest energy binding pose of Droxidopa in the flexible receptor model is associated with an average BFE of −14.0 kcal/mol, corresponding to a pK_d_ of 10.3 (Table 1), and hydrogen bond interactions comparable to those observed in the best binding conformation of the ligand in a rigid β_2_AR model (Figure 11A). Moreover, this structure is characterized by a hydrogen bond network showing an additional polar interaction with Asn312 in the Asp113/Asn312 anchor site (Figure 11B and Table 2). Additionally, this best binding pose of Droxidopa is likely a representative ligand’s conformation because its orientation is similar to that of catecholamines in β_2_AR models reported in the literature [76,86].

In analogy with L-DOPA, the lowest energy Vina conformations of Droxidopa docked into rigid and flexible β_2_AR models displayed less hydrogen bond contacts and average BFEs of −7.5 and −7.9 kcal/mol, respectively, corresponding to binding affinities similar to the rigid AD4 value and comparable with noradrenaline computational and experimental results (Appendix A) [75,76,86,88]. These results implicate that hydrophobic interactions could also play a crucial role in the stability of this catecholamine.

### 3.5. Hydrogen Bonds in β_2_AR-Adrenaline Complexes

The lowest energy conformation of adrenaline from the docking into a rigid β_2_AR model displayed a hydrogen bond network essentially similar to the one of the ethanolamine tail of the ligand in the X-ray crystal structure (Figure 9 and Table 2). The ethanolamine moiety of established hydrogen bond interactions with both anchor site amino acids Asp113 and Asn312, is in good agreement with experimental and computational findings [28,86,90]. Moreover, hydrogen bond interactions were observed between the Asn312 side chain and the β-OH of the ligand, in good agreement with biochemical results [91]. Furthermore, the H-bond distance between Tyr316 and β-OH of adrenaline present in the X-ray crystal structure was partially disrupted and increased from 3.5 to 3.9 Å in the redocked adrenaline. Additionally, Tyr316 was still found to be involved in the formation of hydrogen bonds with the N amino of adrenaline. The Asn293 side chain was located too far away from acceptor and donor atoms of the catecholamine to form hydrogen bonds, which were previously proposed and reported for β_2_AR agonists experimentally and computationally [75,86,88,92,93]. At the same time, this conformational model of the β_2_AR-adrenaline complex presented marked differences in the ligand catechol head of the ligand. Hydrogen bonds of catechol hydroxyls with side chains of serines of the TM-V helical domain are considered the most specific interactions for β_2_AR agonists [75,88,94,95,96,97]. In particular, the H-bonding of both meta-OH and para-OH hydroxyl groups with both Ser203 and Ser207 side chains was not reproduced as in the X-ray crystal structure, the only exception being represented by a decrease of the donor-acceptor Ser207-para-OH distance from 3.5 to 2.6 Å (Table 2). While the distance between hydroxyl groups of the catechol head and the Ser204 side chain increased, additional hydrogen bond interactions arose between both meta-OH and para-OH with Thr118 side chain (Table 2). The involvement of Thr118 in the β_2_AR hydrogen bond network has been recently found in wild type and mutated (T164I) forms of the receptor bound to salbutamol using molecular docking and MD simulations [98]. Similar hydrogen bond interactions were essentially observed for the neutral form of the ligand, the only exception being the loss of the Tyr316-N-amino H-bond [76]. Moreover, the presence of hydrogen bond interactions between residue Thr118 and different β_2_AR ligands has previously been reported in MD simulation studies [98,99,100,101,102].

It is worth noting that adrenaline formed more stable hydrogen bonds on average, as evidenced by shorter distances reported in Table 2, with both its ethanolamine tail and catechol head when docked into a flexible β_2_AR model as compared to conformations of the ligand bound to a rigid model both to the X-ray crystal structure of the receptor. Moreover, two additional hydrogen bonds were found between the catechol head hydroxyls and serines S203 and S207, namely Ser203-para-OH and Ser207-meta-OH, which contributed to the observed dramatic decrease of the BFE (Table 1). The hydrogen bond between the meta hydroxyl group of the catechol ring of adrenaline and residue Asn293 of β_2_AR reported in the X-ray crystal structure (PDB ID: 4LDO) was not reproduced in the lowest energy binding poses of protonated and neutral adrenaline docked in both receptor models [28,76].

### 3.6. Hydrogen Bonds in β_2_AR in Complex with L-DOPA and Droxidopa

The docking of L-DOPA and Droxidopa in both flexible and rigid β_2_AR models produced ligand conformations similar to the best binding poses of adrenaline. It is worth noting that the hydrogen bonds of both L-DOPA and Droxidopa docked in different β_2_AR models displayed some similarities and remarkable differences with the network of polar interactions of the endogenous catecholamine (Figure 10, Figure 11, Figure 12A and Table 2).

The main similarity with polar interactions observed for the endogenous ligand was represented by a reduction in hydrogen bond distances when using flexible receptor models (Table 2). Both ligands formed hydrogen bonds with anchor sites Asp113 and Asn312 through their ethanolamine moieties. In Droxidopa, as observed for adrenaline, the presence of the β-OH group in the ethanolamine tail of the ligand strengthened these polar interactions, resulting in the formation of H-bonds with Asp113 and Asn312 residues in both rigid and flexible β_2_AR models (Table 2). In both receptor models, L-DOPA also showed a hydrogen bond interaction with Asn312 through one oxygen of its carboxylic acid moiety. A similar polar interaction was only present in Droxidopa docked to a flexible β_2_AR receptor (Table 2). In analogy with adrenaline and despite the receptor model, the ethanolamine N of both exogenous ligands developed hydrogen bonds with Tyr316, which also behaved as an anchor site interacting with the β-OH group of Droxidopa docked only in the rigid receptor model (Table 2).

In the rigid receptor, L-DOPA displayed hydrogen bond interactions with β_2_AR residues Ser203 and Ser207 through its catechol moiety para-OH and meta-OH hydroxyls, respectively (Table 2). These polar interactions are different from those reported by experimental and computational studies that predict the formation of hydrogen bonds of meta-OH and para-OH hydroxyl groups of the catechol head of various catecholamines with Ser203 and Ser 207, respectively [86,95,97]. It is interesting to note that the docking of L-DOPA to a rigid β_2_AR model also leads to hydrogen bonding with Ser204 through the para-OH of its catechol head, displaying an additional polar interaction not observed in experiments and computational investigations [86,95,97]. However, the hydrogen bond interaction of the para-OH hydroxyl group of L-DOPA’s catechol with Ser 207 is observed when the exogenous ligand is docked into a flexible β_2_AR model, in good agreement with previous experimental and computational studies [86,95,97] (Table 2). In addition, in the flexible β_2_AR model, both hydroxyl groups of L-DOPA’s catechol head formed hydrogen bonds with Thr118, as observed for the endogenous catecholamine (Table 2). As opposed to L-DOPA, the catechol moiety of Droxidopa was involved in hydrogen bonding only with β_2_AR residues Ser207 and Thr118 in both receptor models (Table 2). In the case of Droxidopa, the para-OH-Ser207 hydrogen bond was found to be particularly stable in rigid and flexible β_2_AR models. Furthermore, both hydroxyl groups of the catechol moiety of Droxidopa formed hydrogen bonds with Thr118 in the flexible β_2_AR model (Table 2).

Similar to the endogenous catecholamine, both exogenous ligands did not display hydrogen bonds with Asn293, which has been found to form a stabilizing polar interaction with adrenaline bound to β_2_AR using different experimental techniques [28,93,96]. In our recent work, we performed the shifting of the grid box center in order to observe the formation of this hydrogen in both endogenous and exogenous ligands [76]. In particular, AD4 lowest energy conformations of L-DOPA and Droxidopa obtained with the shifted grid box center did not exhibit hydrogen bond interactions of the catechol head of the ligand with residue Asn293 of flexible β_2_AR systems, while both ligands showed polar interactions with Thr118 [76]. As observed with adrenaline, the hydrogen bond with residue Asn293 of β_2_AR was instead formed in both L-DOPA and Droxidopa lowest energy binding poses docked into flexible β_2_AR models obtained by Vina calculations [76].

### 3.7. Hydrophobic Contacts of β_2_AR Agonists

The adrenaline ligand exhibited hydrophobic contacts with Val117 and Phe289 residues when docked to a rigid β_2_AR model (Figure 12B). Phe289 was also reported to be a key residue in the formation of hydrophobic interactions with different catecholamines in a previous computational study [86]. When adrenaline was docked in a flexible β_2_AR model, the endogenous ligand also exhibited hydrophobic interactions with Phe290 (Figure 12B). We recently found that neutral adrenaline displayed similar hydrophobic contacts with Val117 and Phe289 residues in rigid β_2_AR models subjected to AD4 and Vina calculations [76]. In flexible β_2_AR models, neutral adrenaline presented the same apolar contacts generated by Vina docking runs of the ligand docked in a rigid receptor, while AD4 calculations showed hydrophobic interactions with Val114 and Val117 residues [76].

Interestingly, L-DOPA exhibited hydrophobic contacts with β_2_AR residues Val114 and Phe193 in rigid receptor models (Figure 12B). It is worth noting that the Vina L-DOPA’s lowest energy conformation in a rigid β_2_AR model also displayed an additional apolar interaction with residue Val117 [76]. In flexible β_2_AR models, the lowest energy binding poses of L-DOPA displayed hydrophobic contacts with Val117 and Phe289 (Figure 12B and Figure 13A). In our recent work, the Vina best binding poses showed the same hydrophobic contacts of the rigid docking approach, namely with Val114, Val117 and Phe193 residues [76].

In rigid β_2_AR models, the lowest energy binding pose of Droxidopa formed hydrophobic interactions with residues Val117 and Phe289 (Figure 12B). We have also recently observed that the Vina best binding pose exhibited apolar contacts with Val114, Val117 and Phe193, as noted for L-DOPA [76]. When β_2_AR binding pocket residues were flexible, the Droxidopa lowest energy conformation displayed hydrophobic contacts with Val114, Val117 and Phe289 (Figure 12B and Figure 13B). In our recent study, the Vina best binding pose of Droxidopa showed more apolar interactions with amino acids Val114, Phe193 and Phe289 [76].

### 3.8. Protein–Ligand Interactions in MD Simulations of β_2_AR-Catecholamine Complexes

Additionally, we performed one microsecond-long AA MD simulations of β_2_AR-catecholamine complexes embedded in POPC:CHOL lipid membranes, which allowed us to determine whether or not the lowest energy binding poses were stable. After 300 ns, the receptor C-alpha atoms RMSDs reached a plateau value of less than 4 Å in the β_2_AR-adrenaline complex, indicating that the endogenous ligand provided structural stability to the entire protein (Figure 14A). This stability persisted until 700 ns into the simulation period, when there were fewer changes in protein conformation. When compared to β_2_AR adreanline’s lower RMSD values and stability at 300 ns, Droxidopa and L-Dopa show less fluctuation after 300ns, suggesting that these exogenous ligands might induce more conformational changes within the protein than adrenaline (Figure 14A). The RMSDs of the ligand’s heavy atoms are shown in Figure 14B. It is worth noting that the stability of each ligand is achieved after 300 ns of the simulation time, which is in line with prior findings. The conformational variations in L-Dopa and Droxidopa were found to be larger than in the β_2_AR-adrenaline system; however, after 300 ns, all simulations achieved stability with less changes in the conformational state of each exogenous ligand.

The hydrogen bond network of L-DOPA and Droxidopa with β_2_AR residues during MD simulations was very similar to the one observed with the CHARMM FF (Figure 15) and comparable to that of adrenaline (Appendix A), supporting the results obtained with molecular docking calculations.

The contact maps illustrated in Figure 16 for the L-Dopa and Droxidopa molecules indicate excellent interactions between ligands and the β_2_AR residues. In the green to red shades, while analyzing the 1000 ns simulation, we observe that the L-Dopa forms stronger connections with V114, F193, F289, F290 and N293 and loose or fading interactions with D113, V117, S203, S204, S207, Y308 and N312, which may also include the hydrophobic interaction component of the binding site. When analyzing the Droxidopa, we found that it forms strong connections with D113, V114, V117, F289, F290, N312 and Y316, and fading bonds with T118, S203, S204, S207, W286 and N293 throughout the 1000 ns simulation time. In addition, Droxidopa has a greater affinity for binding than L-Dopa, which is convincing evidence that it may be more effective. We observed that Adrenaline’s probability contact map with β_2_AR residues is comparable to L-Dopa’s (Appendix A), displaying excellent contacts with D113, V114, V117, F193, F289 and N312 and fading bonds for T110, T118, S203, S207, F290, N293, Y308 and Y316. Furthermore, we can see that Adrenaline’s interaction profile is extremely comparable to that of L-Dopa, while that of Droxidopa’s interaction profile is rather higher. BFE calculations using the MM-PBSA method indicate this as well. Droxidopa has a stronger interaction with the β_2_AR receptor than L-Dopa or Adrenaline, as evidenced by its BFE reported in Table 1, indicating that all three interaction profiles and residue characterizations can be trusted.

Interestingly, MM-PBSA calculations over a 1 μs MD simulation of the β_2_AR-adrenaline complex yielded a BFE of −4.5 +/− 0.4 kcal/mol (pK_d_ = 3.3), which was higher than the experimental value of the endogenous ligand (−8.2 kcal/mol pK_d_ = 6.1–6.5) (Table 1). More interestingly, in the simulated receptor–ligand complex, L-DOPA displayed a BFE of −5.4 +/− 0.4 kcal/mol (pK_d_ = 3.9), which was lower than the adrenaline ligand value (Table 1). Furthermore, MD simulations of the β_2_AR-Droxidopa complex exhibited a catecholamine BFE of −12.5 +/−0.4 kcal/ mol, corresponding to a pK_d_ of 9.3 (Table 1), better than noradrenaline experimental values (i.e., 5.0 and 5.4). On the contrary, the adrenaline and L-DOPA BFEs estimated for each ligand parameterized with the OPLS FF were more positive than not only those obtained by both rigid and flexible docking calculations, but also those from MD simulations performed with the CHARMM FF [76], suggesting that this FF might better reproduce the conformational space of these ligands.

We examined the contributions of β_2_AR residues to the BFE of each catecholamine, in order to better understand protein–ligand interactions (Figure 17). For the most part, the hydrogen bond and hydrophobic interactions of the receptor residues (particularly D113, Y308, N312, S203, S204, S207, T118, N293, V114, V117 and F193) were shown to have a positive effect on adrenaline’s BFE, confirming both experimental and computational results [75,76,86,88,94,97]. Intriguingly, six negatively charged amino acids (E107, E180, E188, D192, D300, E306) and two positively charged residues (R175 and K305) had a stabilizing and destabilizing influence on the β_2_AR-adrenaline complex, despite being positioned on the periphery of the β_2_AR binding pocket. Figure 17 shows that L-DOPA substantially interacted with the same residues as adrenaline and, particularly, with hydrophilic (S203, S204, T118, N293, Y308) and hydrophobic (V114, V117, F193 and F289) residues. Furthermore, we found that D113 had a poor contact with the ligand, which may have been owing to the ligand’s carboxilic group being in an unfavorable position relative to the aspartic acid equivalent moiety. The β_2_AR residues that contributed more to the Droxidopa’s BFE were similar to those found for L-DOPA and adrenaline, and those that bound more strongly to the catecholamine were V114, V117 and F289, in good agreement with the interaction pattern described above and our previous findings with the CHARMM FF [76]. D113, similar to L-DOPA, had a modest interaction with Droxidopa, although its energetic contribution was less favorable.

## 4. Discussion

To study the interaction of the adrenergic receptor β_2_AR with endogenous and exogenous catecholamines, we have recently used molecular docking calculations and MD simulations [76]. Rigid docking calculations produced the lowest energy conformations of all investigated ligands with pK_d_s similar to experimental values. In contrast, all catecholamines docked into flexible β_2_AR models had pK_d_s exceeding experimental values, which were also confirmed by BFEs obtained from MM-PBSA calculations performed on MD simulations of β_2_AR-catecholamine complexes.

In particular, the molecular docking of endogenous ligands, such as adrenaline, noradrenaline and dopamine, into both rigid and flexible β_2_AR models led to the discovery of novel binding poses. These new conformations differ from experimental and computationally anticipated structures, indicating that catecholamines can adopt more energetically stable binding modes. The best binding poses of exogenous ligands L-DOPA and Droxidopa were similar to endogenous catecholamines dopamine and noradrenaline, respectively.

Thus, to further investigate the aforementioned binding poses of endogenous and exogenous ligands through MD simulations, we parameterized the OPLS AA FFs of these small organic molecules. The JOYCE FF functions accurately reproduce the QM torsional profiles of each dihedral angle of the zwitterionic L-DOPA. These dihedral torsional profiles resemble those of zwitterionic L-Tyrosine, as reported by Del Galdo et al., in 2018 [85]. The agreement between QM and JOYCE torsional profiles for charged groups in zwitterionic condition was better than that found by Del Galdo et al., in 2018 [85] for the comparable amino acid L-Tyrosine, confirming the excellent quality of the FF parameterization of L-DOPA. Similar to L-DOPA, the QM torsional profiles of zwitterionic Droxidopa dihedrals matched JOYCE FF characteristics. The QM and JOYCE torsional profiles of Droxidopa dihedrals are very close to those of L-DOPA. In addition, QM and JOYCE torsional profiles of protonated adrenaline were similar and in good agreement with those observed for the two exogenous ligands.

To evaluate the L-DOPA and Droxidopa FFs, we performed MD simulations of each ligand in both vacuum and aqueous solution. Notably, both L-DOPA and Droxidopa showed similar dihedral distributions in vacuum and water for each dihedral angle. The Helmholtz free energy variation patterns were identical to the QM energy profiles for both L-DOPA and Droxidopa. Each profile created using the JOYCE program as a statistical ensemble is compared to the OPLS AA FF profile. The entropic contribution is only present in the free energy, as reported by Del Galdo et al., in 2018 [85] for zwitterionic Tyrosine. Despite this difference, the comparison has allowed us to qualitatively assess the accuracy of the developed FFs for various dihedral angles of each ligand.

Since adrenaline is predominantly protonated at a physiological pH, we initially employed this form of the ligand to perform its redocking into rigid and flexible β_2_AR models, yielding conformations distinct from that of the X-ray crystal structure (ligand heavy atom RMSDs of 2.2 and 2.4 Å, respectively) (Figure 9) [28]. While the ethanolamine tail of the redocked adrenaline binds to anchor site amino acids Asp113/Asn312, the catechol head interacts with different residues in the crystal and redocked structures, respectively. According to experimental studies, the BFE of adrenaline varies from −8.3 kcal/mol to −8.9 kcal/mol, showing that the rigid redocking structure of adrenaline is almost in its more energetically stable conformation (BFE = −8.1 kcal/mol). The lowest energy conformation of adrenaline in flexible receptor models is not only more stable than the rigid docking conformation, but also has a pK_d_ better than the experimental values reported in the literature (Table 1) [75,88]. This remarkable result is most probably a consequence of the larger conformational space sampled by the flexible redocking as compared to the rigid docking calculation [76]. Taking into account the variety of β_2_AR active states in physiological [103] and tumoral [104] conditions, it is plausible to predict that the endogenous catecholamine could interact with the receptor adopting conformations more similar to those described by flexible docking calculations and MD simulations.

Interestingly, the docking of L-DOPA to a rigid receptor model generated best binding poses unable to create hydrogen bonds in an adrenaline-like manner. As a consequence, the average BFE of L-DOPA docked into a rigid β_2_AR model was −6.4 kcal/mol, higher than the Vina finding (−7.3 kcal/mol), corresponding to a pK_d_ of 4.7 (Appendix A and Table 1) [76]. The docking of L-DOPA into the flexible β_2_AR model induced more ligand–receptor hydrogen bond interactions than docking into the rigid receptor model. The ligand’s catechol head formed hydrogen bonds with the binding site residue Thr118. Moreover, L-DOPA docked into a flexible β_2_AR model has a pK_d_ of 9.4, which is comparable to the value observed for adrenaline (9.2) (Table 1). The lowest energy conformation of Droxidopa bound to a rigid receptor showed a BFE of −7.4 kcal/mol, which is comparable to the one we observed for noradrenaline (−7.6 kcal/mol) [75,76,88], demonstrating the validity of the molecular docking protocol. Droxidopa, similar to L-DOPA, docked into the flexible β_2_AR model resulted in lowest energy conformations having stronger hydrogen bond interactions with both anchor sites of the binding pocket of the receptor (Table 2). Droxidopa’s lowest energy conformation in the flexible receptor model had a BFE of −14.0 kcal/mol (pK_d_ = 10.3), polar interactions similar to the ligand’s best binding pose in the rigid β_2_AR model (Table 1 and Table 2) and an orientation similar to that of catecholamines in β_2_AR models [86]. Although the contribution of hydrophobic interactions has to be taken into account, this favorable BFE is due to a hydrogen bond network larger than those observed for adrenaline and L-DOPA [76].

When docked into a rigid β_2_AR model, the lowest energy conformation of adrenaline revealed a hydrogen bond network similar to that of the ligand’s ethanolamine tail in the X-ray crystal structure, showing polar interactions with Asp113 and Asn312. In particular, the Asn312 side chain formed hydrogen bonds with the ligand’s β-OH, as expected biochemically [91]. Tyr316 was also involved in the formation of hydrogen bonds with adrenaline’s ethanolamine tail (Figure 9 and Table 2). However, the most specific polar interactions for β_2_AR agonists are hydrogen bonds between catechol hydroxyls and serines in the TM-V helical domain [94,95,96]. We only observed the conservation of the hydrogen bond between S207 and the para-OH of the ligand, as compared to the β_2_AR X-ray structure (Table 2). Interestingly, both catechol hydroxyls formed polar interactions with Thr118. Molecular docking and MD simulations have recently revealed the role of Thr118 in the β_2_AR hydrogen bond network [76]. Furthermore, MD simulations studies have reported hydrogen bond interactions between residue Thr118 and several β_2_AR ligands [98,99,100,101,102]. When docked into a flexible β_2_AR model, adrenaline established more stable hydrogen bonds with its ethanolamine tail and catechol head, as compared to conformations of the ligand bound to a rigid receptor model or the X-ray crystal structure (Figure 9 and Table 2). We observed the formation of additional hydrogen bonds between catechol head hydroxyls and serines S203 and S207 (Ser203-para-OH and Ser207-meta-OH), which contributed to the BFE decrease. The Asn293 side chain did not form hydrogen bonds with the meta-OH of adrenaline, as in Ring et al.’s X-ray structure [28] and a previous computational study of β_2_AR agonists [86].

L-DOPA and Droxidopa conformations were similar to adrenaline’s best binding poses in both flexible and rigid β_2_AR models. The tail moieties of both ligands formed hydrogen bond interactions with anchor sites Asp113 and Asn312 (Figure 10, Figure 11 and Table 2). In the rigid β_2_AR model, the catechol head of L-DOPA formed hydrogen bonds with Ser203 and Ser207 through its para- and meta-OH hydroxyl groups, respectively, in good agreement with experimental and computational results [86,94,95,96]. Additionally, L-DOPA docked into a rigid receptor model formed hydrogen bonds with Ser204 through the para-OH of its catechol moiety, an interesting polar interaction not supported by previous experimental and computational investigations (Table 2). Both hydroxyl groups of L-DOPA and Droxidopa catechol heads formed hydrogen bonds with Thr118 in flexible β_2_AR models. In both receptor models, Droxidopa’s catechol head also displayed polar interactions with Ser207, as shown by the quite stable para-OH-Ser207 hydrogen bond (Figure 11 and Table 2). Although we shifted the grid box center toward Asn293 of flexible β_2_AR systems [76], the catechol heads of both L-DOPA and Droxidopa did not form hydrogen bonds with Asn293 and displayed polar interactions with Thr118 in AD4 lowest energy conformations. Interestingly, the L-DOPA and Droxidopa lowest energy binding poses docked into flexible β_2_AR models produced by Vina simulations created a hydrogen bond with Asn293 of β_2_AR [76]. It is worth noting that the hydrogen bond network of L-DOPA and Droxidopa observed through molecular docking calculations was also reproduced during MD simulations of β_2_AR-catecholamine complexes performed with CHARMM [76] and OPLS AA FFs (Figure 15, Appendix A). The OPLS AA FF of both ligands were less successful in keeping stable hydrogen bonds with Ser203 and Ser207, but generated similar percentages of hydrogen bond formation with other key amino acids, such as Asp113, N312, Thr118 and Y316, especially in the case of Droxidopa, as compared to corresponding results obtained with the CHARMM AA FF (Figure 15 and Appendix A) [76].

In rigid and flexible β_2_AR models, adrenaline showed hydrophobic interactions with Val117 and Phe289 residues. In recent computational studies, Phe289 has been found to be a critical residue in the formation of hydrophobic interactions with catecholamines [76,86]. However, adrenaline docked in a flexible β_2_AR model also revealed hydrophobic interactions with Phe290. Molecular docking calculations employing rigid receptor models showed that L-DOPA had hydrophobic interactions with β_2_AR residues Val114 and Phe193. When L-DOPA was docked into flexible β_2_AR models, the exogenous ligand formed apolar interactions with Val117 and Phe289, the same residues were observed for the endogenous ligand. Similarly to adrenaline, Droxidopa’s lowest energy conformation interacted with Val117 and Phe289 in rigid β_2_AR models. When β_2_AR binding pocket residues were flexible, the Droxidopa best binding pose exhibited hydrophobic contacts with Val114, Val117 and Phe289. It is remarkable that MD simulations of β_2_AR-catecholamine complexes performed with CHARMM and OPLS AA FFs displayed the formation of hydrophobic interactions with the same amino acids obtained by molecular docking calculations (Figure 16, Figure 17, Appendix A), confirming the quality of both ligands’ FFs and their ability to reproduce these important apolar interactions [76]. Due to the broader conformational space sampled by MD simulations, this methodology usually provides better and more reliable binding poses of ligands than lowest energy conformations obtained by flexible and rigid molecular docking using both AD4 and Vina calculations.

In order to validate the hypothesis that drugs employed in the treatment of PD could display binding conformations similar to those observed for L-DOPA and Droxidopa, we also performed the molecular docking of carbidopa, foslevodopa and foscarbidopa to a rigid β_2_AR model using the latest version of Vina (v. 1.2.3) (Appendix A) [105]. The lowest energy conformations of carbidopa, foslevodopa and foscarbidopa are structurally similar to those observed for adrenaline, L-DOPA and Droxidopa, respectively, displaying BFEs of −7.5, −6.4 and −7.2 kcal/mol, respectively, comparable with the values obtained for the endogenous and exogenous ligands. Moreover, the hydrogen bond network of these drugs is also characterized by polar interactions with the same amino acids, including Thr118, observed in best binding poses of adrenaline, L-DOPA and Droxidopa (Appendix A). The main motivation of our study was to provide a step-by-step perspective guide for the investigation of β_2_AR-catecholamine complexes, including the development of FF parameters of ligands, molecular docking calculations of each ligand to both flexible and rigid receptor models and MD simulations of lowest energy binding poses of ligands docked to protein models embedded in lipid membranes. Future objectives will include the FF parameterization of additional related drugs (see above) and MD simulations of β_2_AR-drug complexes embedded in membranes with more physiological lipid composition at atomistic and coarse grained levels.

## 5. Conclusions

OPLS AA FFs have been developed and tested for protonated adrenaline, and zwitterionic L-DOPA and Droxidopa molecules by performing QM calculations and MD simulations in water. QM torsional profiles of all ligands resembled those of zwitterionic L-Tyrosine [85]. Moreover, QM and JOYCE torsional profiles of α, β, γ, δ, ϵ and η dihedrals of Droxidopa were similar to those observed for L-DOPA.

Molecular docking calculations have highlighted that the binding of exogenous catecholamines to β_2_AR is favored by the formation of hydrogen bonds between their catechol head and ethanolamine tail moieties and key amino acids, such as Ser203, Ser204, Ser207, Asp113 and Asn312, as reported in previous experimental and computational studies [75,86,88,94,97]. This network of polar interactions involves Tyr316 and Thr118 residues of β_2_AR and ethanolamine and catechol moieties, respectively, increasing the stability of lowest energy binding poses of each investigated ligand. Additionally, each catecholamine also formed hydrophobic interactions with β_2_AR residues Val114, Val117, Phe193, Phe289 and Phe290, previously observed to interact with similar agonists [86].

Furthermore, OPLS AA FF parameters of adrenaline, L-DOPA and Droxidopa were also applied in 1 μs long MD simulations of β_2_AR-catecholamine complexes embedded in lipid membranes. MD simulations of these β_2_AR-catecholamine complexes exhibited hydrogen bond and hydrophobic interactions comparable to those derived from molecular docking calculations and MD simulations performed with the CHARMM AA FF [76].

Since we have recently shown how L-DOPA and Droxidopa drugs interact with β_2_AR [76], our preliminary molecular docking and latest MD simulations findings strengthen the hypothesis that similar binding modes could also be predicted for the other drugs in use for treating patients with Parkinson’s disease. 

## Figures and Tables

**Figure 1 life-12-01393-f001:**
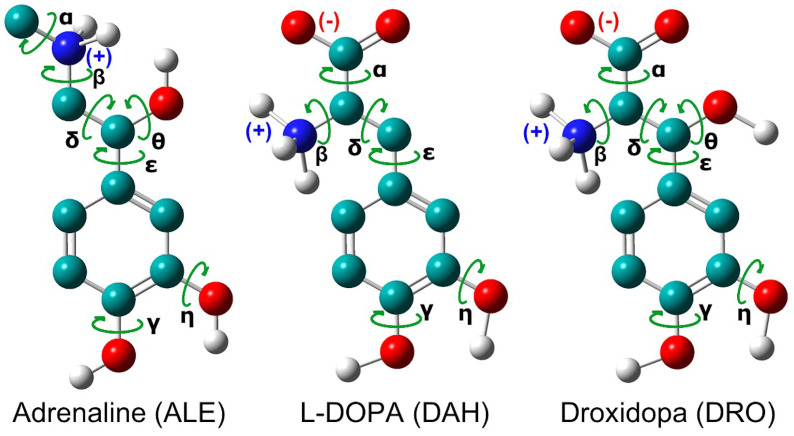
Chemical structures of protonated adrenaline and zwitterionic L-DOPA and Droxidopa ligands. Carbon, hydrogen, nitrogen and oxygen atoms are shown in cyan, white, blue and red, respectively. Non-polar hydrogens are not shown for clarity. Dihedral angles of each ligand are highlighted by green arrows.

**Figure 2 life-12-01393-f002:**
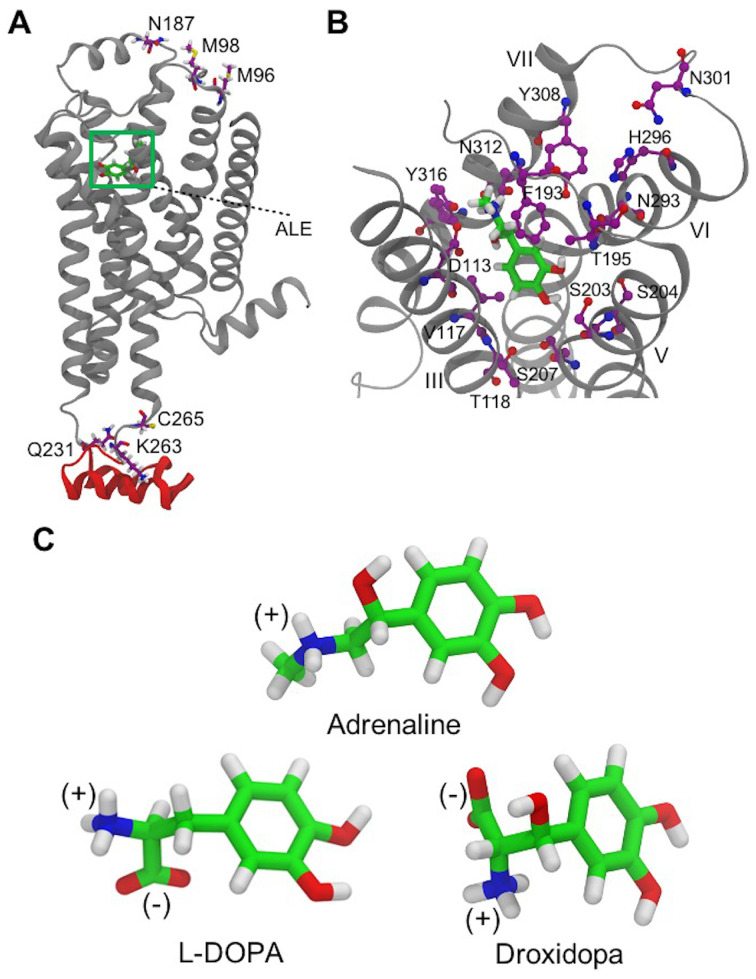
(**A**) A side view of the β_2_AR model shows the main changes applied to the 4LDO original structure with adrenaline (ALE) in its crystallographic binding site highlighted by a green square. β_2_AR TM α-helices and the ICL3 domain are in gray and red, respectively. M96, M98, N187 and C265 residues, which are mutated in the X-ray crystal structure, and Q231 and K263 residues connecting 4LDO to the ICL3 domain are shown in licorice representation. (**B**) β_2_AR residues of the binding pocket interacting with crystallographic adrenaline (green) are shown in purple without hydrogens. TM α-helices III, V, VI and VII are displayed in grey. (**C**) Chemical structures of protonated adrenaline and zwitterionic L-DOPA and Droxidopa molecules employed in the molecular docking to rigid and flexible β_2_AR receptor models.

**Figure 3 life-12-01393-f003:**
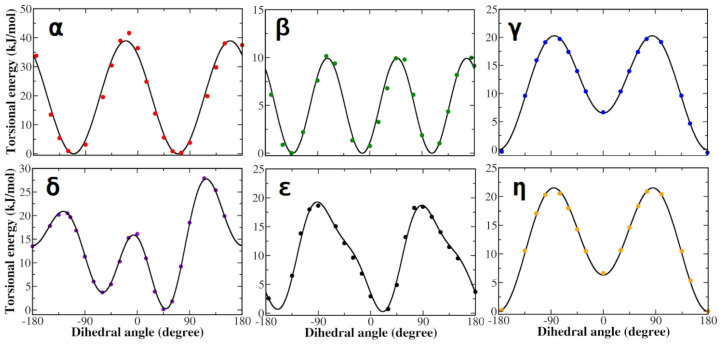
Torsional profiles derived from QM calculations (dots) and JOYCE (lines) for zwitterionic L-DOPA. Each panel refers to dihedral angles defined in Figure 1. Energies are reported in kilojoules per mole (1 kJ/mol = 0.24 kcal/mol).

**Figure 4 life-12-01393-f004:**
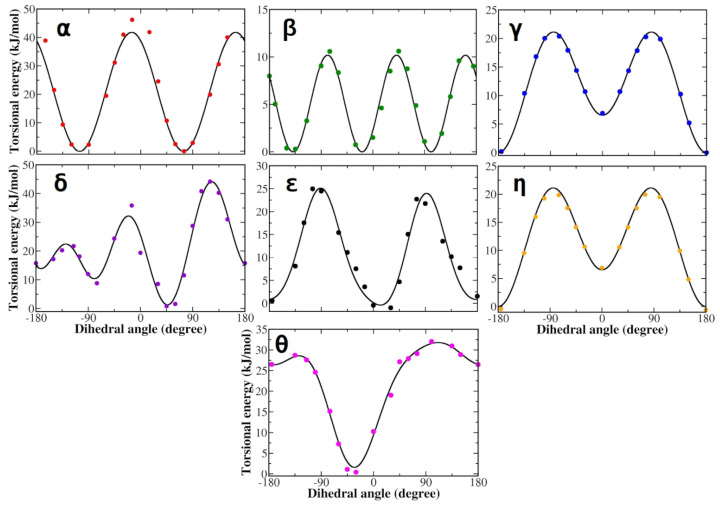
Torsional profiles derived from QM calculations (dots) and JOYCE (lines) for zwitterionic Droxidopa. Each panel refers to one of the dihedral angles defined in Figure 1. Energies are reported in kilojoules per mole (1 kJ/mol = 0.24 kcal/mol).

**Figure 5 life-12-01393-f005:**
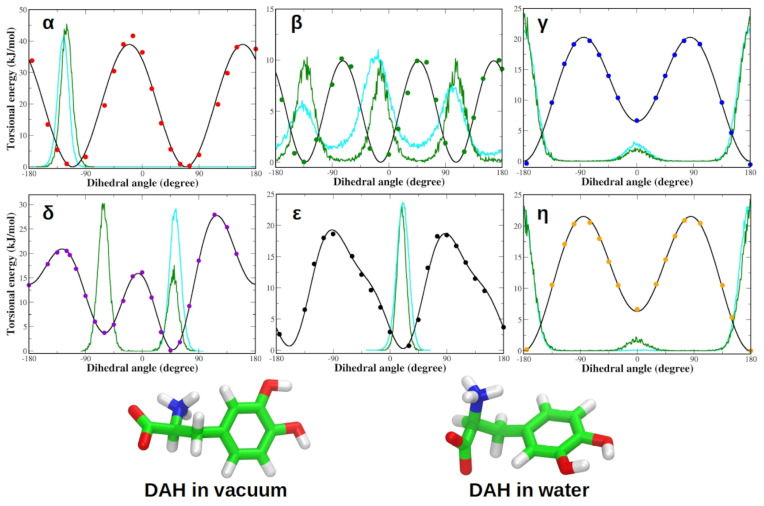
(**Top**) Dihedral distributions in vacuum (green) and water (cyan) are compared to torsional profiles obtained by QM calculations (dots) and JOYCE (lines) for zwitterionic L-DOPA. Each panel refers to one of the dihedral angles defined in Figure 1. (**Bottom**) L-DOPA structures simulated for 20 ns and 100 ns in vacuum and water, respectively. Energies are reported in kilojoules per mole (1 kJ/mol = 0.24 kcal/mol).

**Figure 6 life-12-01393-f006:**
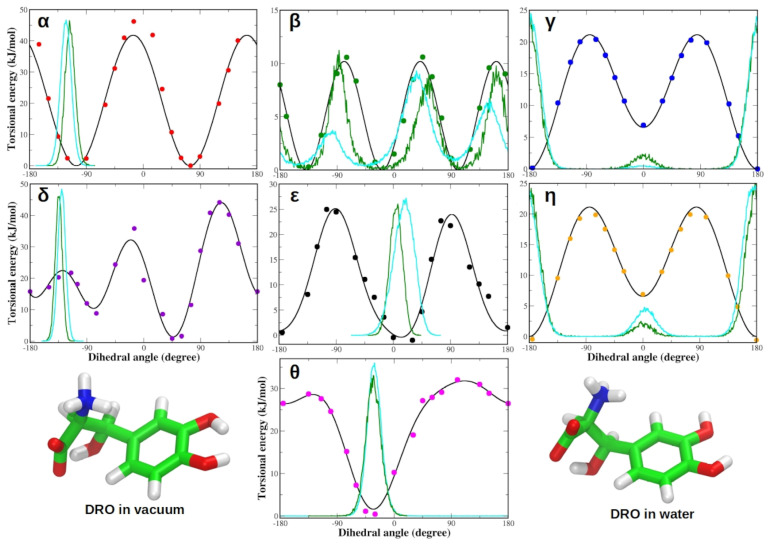
(**Top**) Dihedral distributions in vacuum (green) and water (cyan) compared to torsional profiles obtained by QM calculations (dots) and JOYCE (lines) for zwitterionic Droxidopa. Each panel refers to one of the dihedral angles defined in Figure 1. (**Bottom**) Droxidopa structures simulated for 20 ns and 100 ns in vacuum and water, respectively. Energies are reported in kilojoules per mole (1 kJ/mol = 0.24 kcal/mol).

**Figure 7 life-12-01393-f007:**
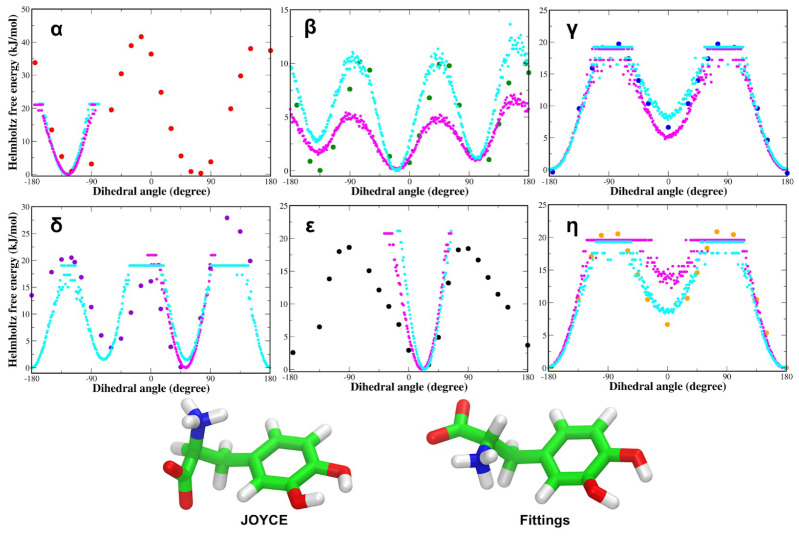
(**Top**) Zwitterionic L-DOPA QM torsional energy profiles (circles) compared with Helmholtz free energy variations as a function of the dihedral angle, as obtained from MD simulations performed using JOYCE (magenta dots) and OPLS (cyan dots) FFs. (**Bottom**) 100 ns structures of L-DOPA from MD simulations using JOYCE and OPLS (Fittings) FF parameters, respectively. Energies are reported in kilojoules per mole (1 kJ/mol = 0.24 kcal/mol).

**Figure 8 life-12-01393-f008:**
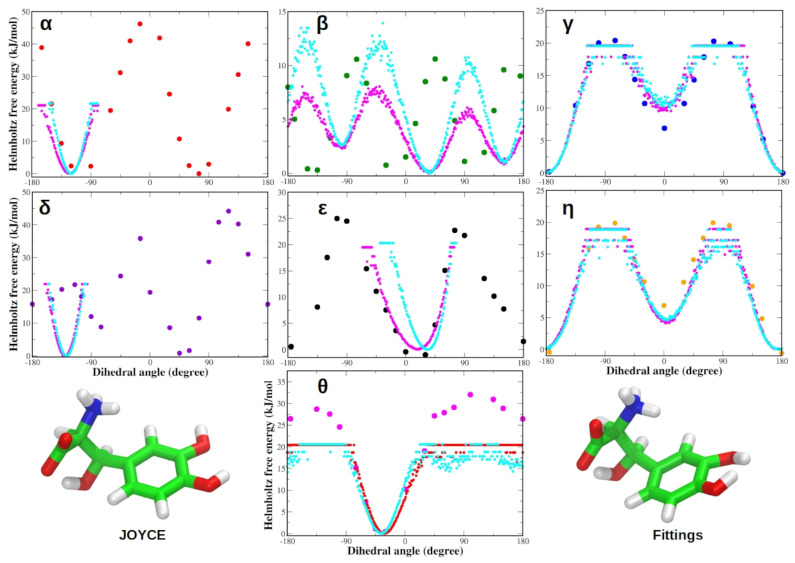
(**Top**) Zwitterionic Droxidopa QM torsional energy profiles (circles) compared with Helmholtz free energy variations as a function of the dihedral angle, as obtained from MD simulations performed using JOYCE (magenta dots; for clarity, red dots are used for the dihedral θ) and OPLS AA FFs (cyan dots); (**Bottom**) 100 ns structures of Droxidopa from MD simulations using JOYCE and OPLS (Fittings) FF parameters, respectively. Energies are reported in kilojoules per mole (1 kJ/mol = 0.24 kcal/mol).

**Figure 9 life-12-01393-f009:**
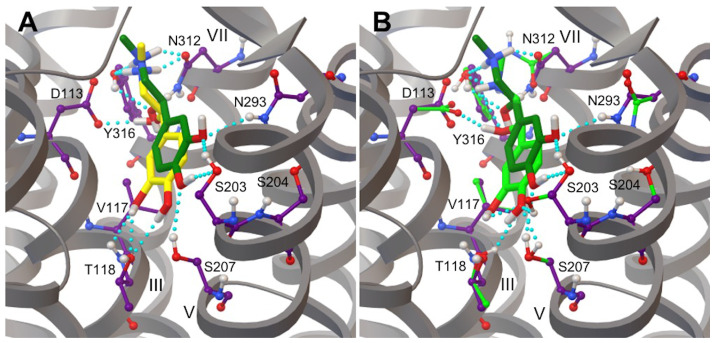
Conformations of adrenaline from the β_2_AR X-ray crystal structure (PDB ID: 4LDO) and the redocking of the ligand to rigid (**A**) and flexible (**B**) models of the receptor are shown in yellow and green, respectively. Lowest energy binding poses of adrenaline from the molecular docking to rigid (yellow) and flexible (green) β_2_AR models are compared to the ligand’s conformation in the X-ray crystal structure (dark green). The protein is shown as grey ribbons and β_2_AR side chains in contact with adrenaline are shown as sticks and balls. Carbon atoms of rigid and flexible side chains of β_2_AR amino acid residues interacting with adrenaline are shown in purple and green, respectively. Hydrogen bonds are shown with cyan spheres. All non-polar hydrogens are not shown.

**Figure 10 life-12-01393-f010:**
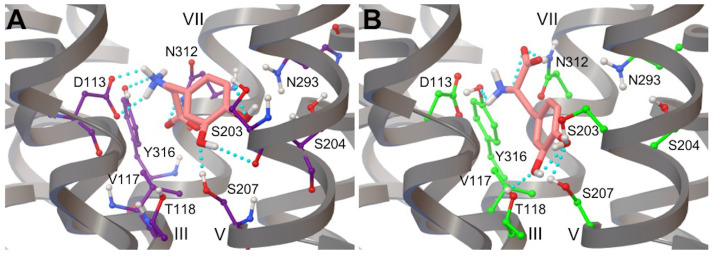
Binding poses of L-DOPA obtained from the molecular docking into (**A**) rigid and (**B**) flexible models of the β_2_AR receptor. The largest cluster conformation of L-DOPA (salmon) in a β_2_AR rigid model displays more hydrogen bonds with anchor sites as compared to the lowest energy binding pose of the ligand in a flexible receptor. Carbon atoms of flexible side chains in the binding site are in green. View point and color code as in Figure 9.

**Figure 11 life-12-01393-f011:**
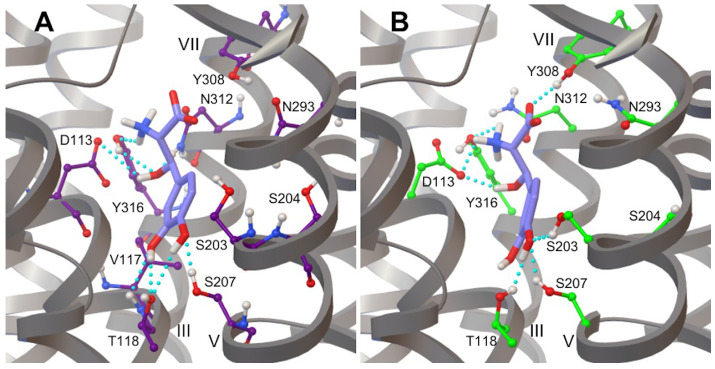
Conformations from the molecular docking of Droxidopa to (**A**) rigid and (**B**) flexible models of the β_2_AR receptor. The lowest energy binding pose of Droxidopa (violet) in a rigid receptor displays more hydrogen bond interactions with anchor sites when compared to the lowest energy binding conformation of the ligand in a flexible β_2_AR model. Carbon atoms of flexible side chains in the binding site are in green. View point and color code as in Figure 9.

**Figure 12 life-12-01393-f012:**
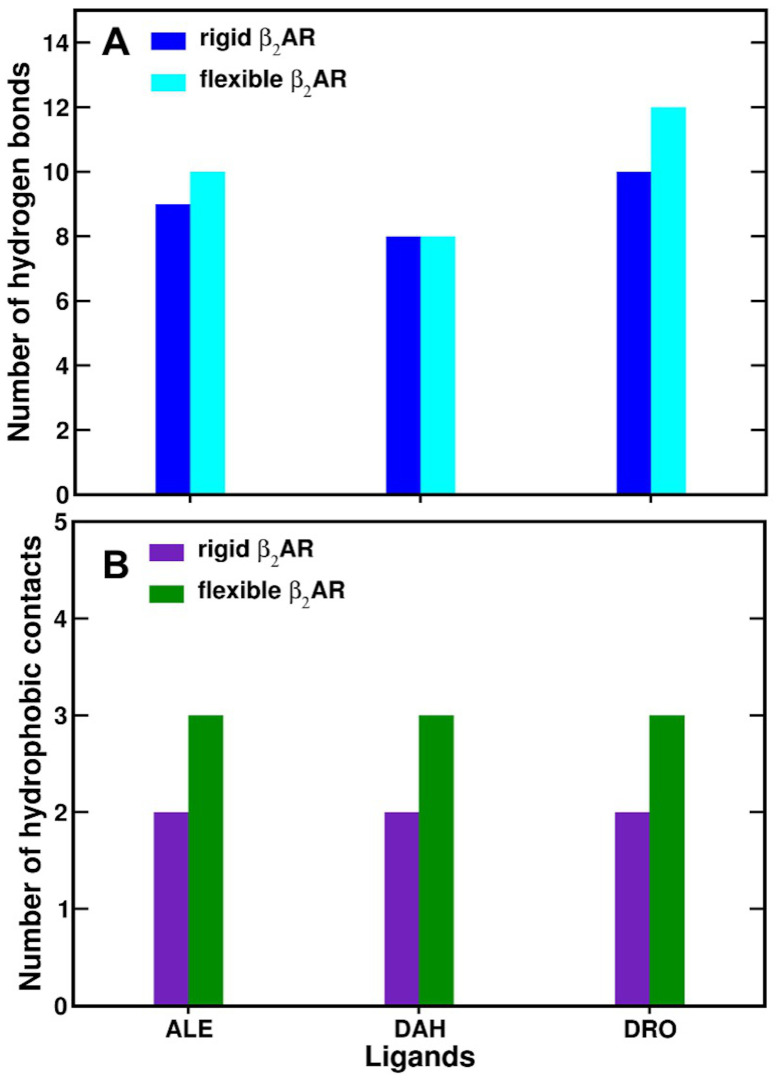
Number of (**A**) hydrogen bond and (**B**) hydrophobic contacts between β_2_AR amino acids and different catecholamines docked into rigid and flexible receptor models obtained by molecular docking calculations.

**Figure 13 life-12-01393-f013:**
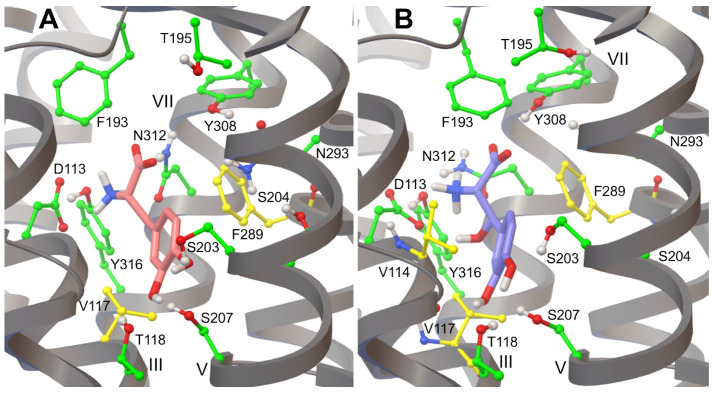
Receptor hydrophobic residues in contact with lowest energy binding poses of (**A**) L-DOPA and (**B**) Droxidopa docked into flexible β_2_AR models are shown in yellow. View point and color code as in Figure 9.

**Figure 14 life-12-01393-f014:**
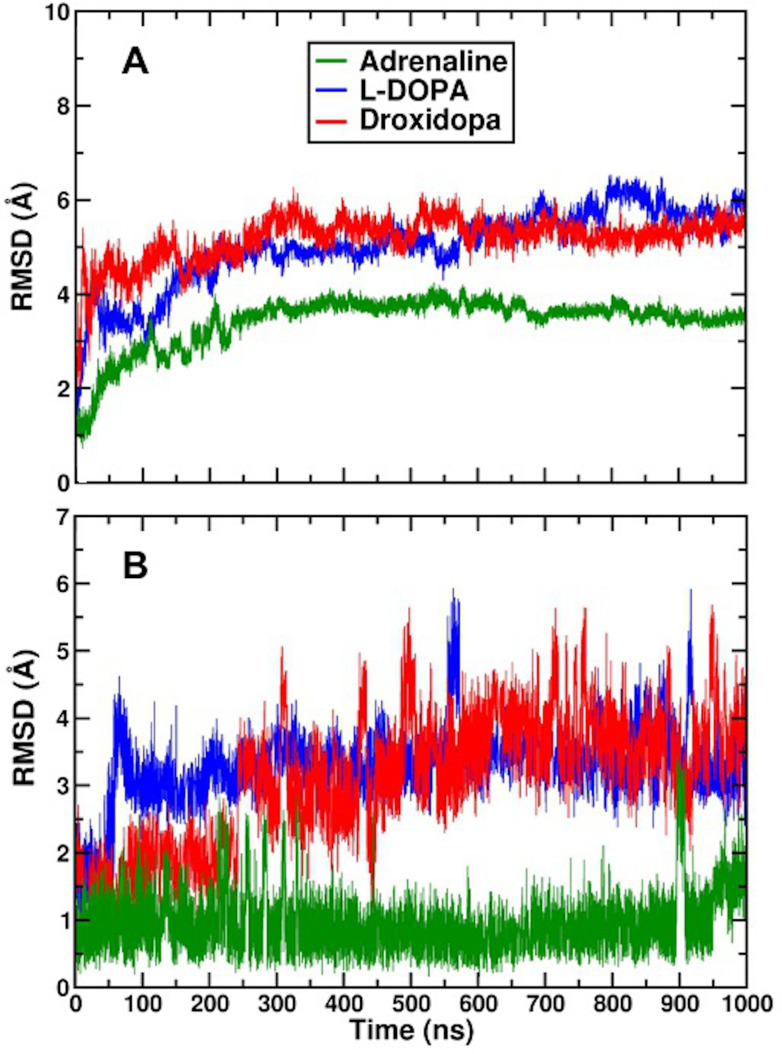
(**A**) RMSDs of β_2_AR Cα atoms in receptor complexes with adrenaline (green), L-DOPA (blue) and Droxidopa (red). (**B**) RMSDs of ligands heavy atoms.

**Figure 15 life-12-01393-f015:**
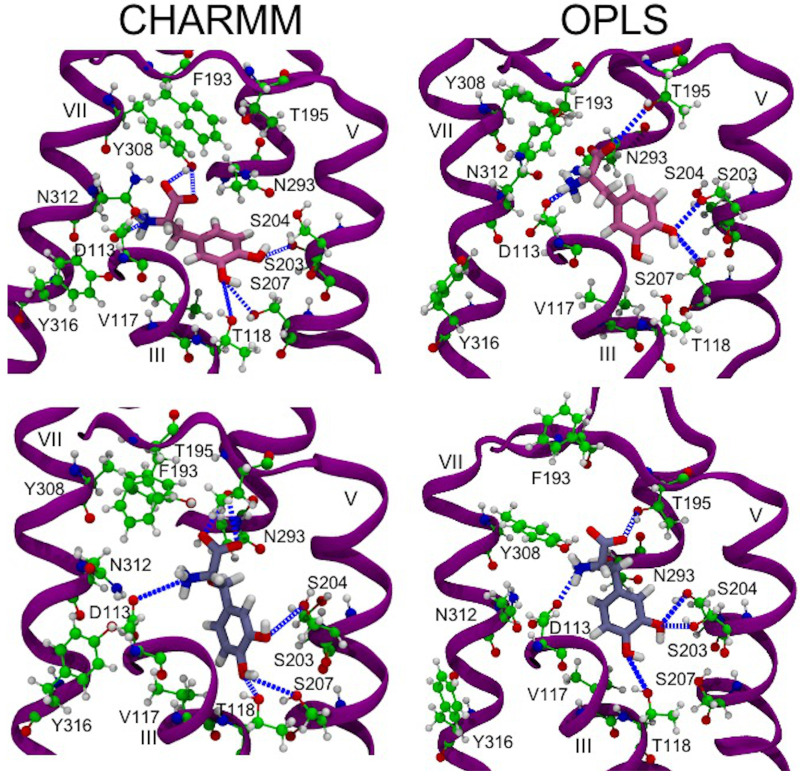
Snapshots of MD simulations of β_2_AR-catecholamine complexes show the hydrogen bond network of (**top**) L-DOPA and (**bottom**) Droxidopa ligands with key amino acids of the receptor binding pocket using CHARMM and OPLS AA FFs. Hydrogen bonds are highlighted with dashed blue lines. View point and color code as in Figure 5 of ref. [76].

**Figure 16 life-12-01393-f016:**
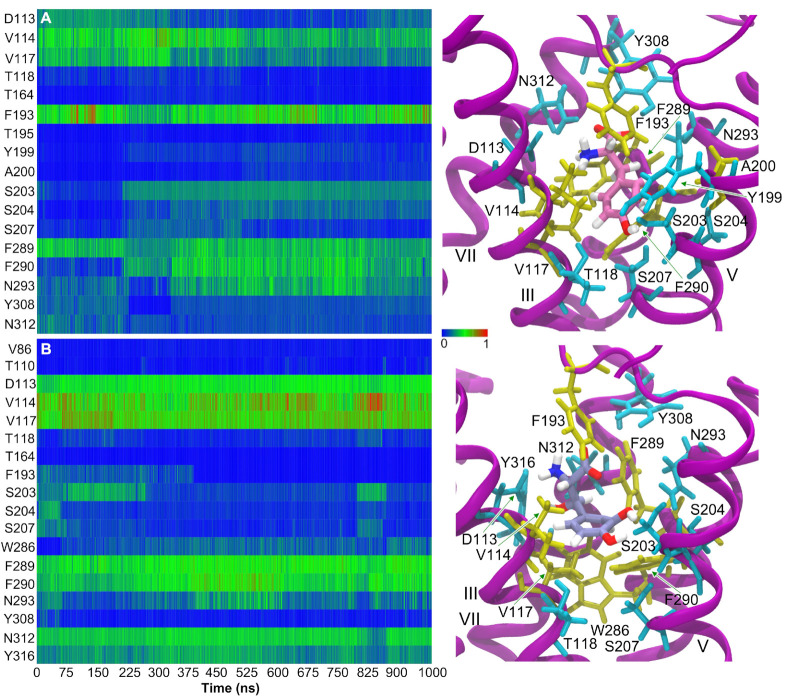
Normalized number of contacts of (**A**) L-DOPA and (**B**) Droxidopa with the receptor residues extracted from 1 μs AA MD simulations of the β_2_AR-catecholamine complexes performed with the OPLS AA FF. Snapshots from MD simulations show β_2_AR hydrophilic (cyan) and hydrophobic (yellow) residues interacting with (**top**) L-DOPA and (**bottom**) Droxidopa ligands. For clarity, only β_2_AR residues displaying the largest percentages of contacts with the ligand are shown. Side chains of some residues are indicated by green arrows. This analysis was performed over the whole 1 μs trajectory.

**Figure 17 life-12-01393-f017:**
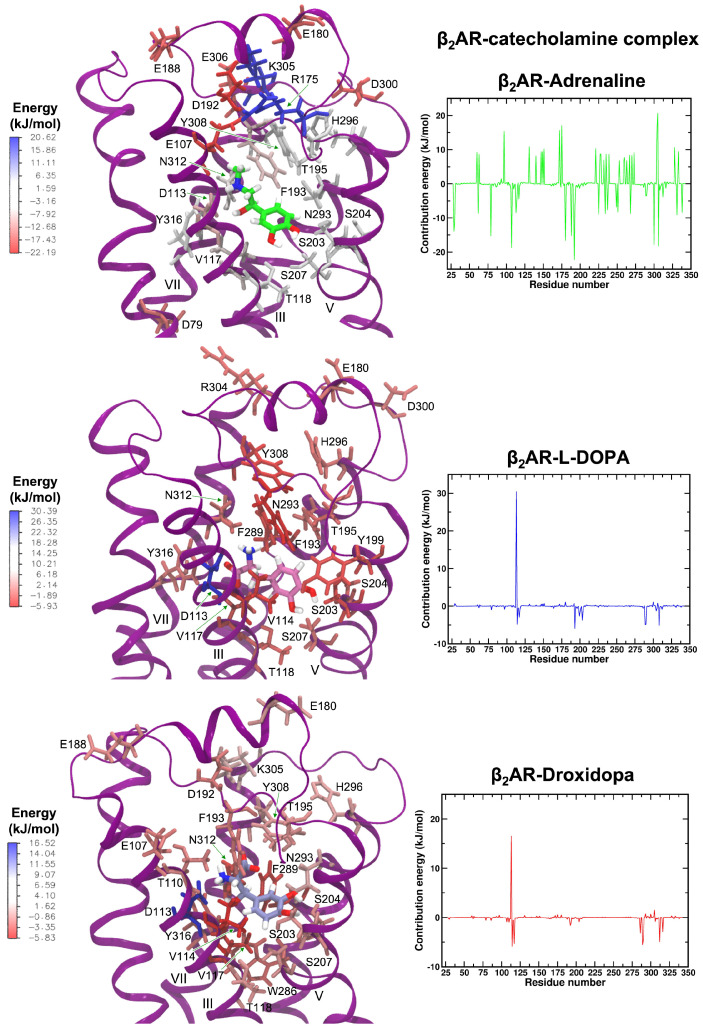
The mapping of energy contributions on the structure of β_2_AR-catecholamine complexes and contribution energies of β_2_AR residues for (**top**) adrenaline, (**middle**) L-DOPA and (**bottom**) Droxidopa ligands. Energies are reported in kilojoules per mole (1 kJ/mol = 0.24 kcal/mol).

**Table 1 life-12-01393-t001:** Calculated binding affinities, pK_d_, of different ligands for rigid and flexible β_2_AR models obtained by molecular docking calculations in comparison with experimental values.

Ligand	Calculated Binding Affinity pK_d_ ^a^		Experimental Binding Affinity pK_d_
	**Rigid Model**	**Flexible Model**	
Adrenaline	5.9	9.2	6.1 ^b^–6.5 ^c^ (3.3) ^d^
L-DOPA	4.7	9.4	NA (3.9)
Droxidopa	5.4	10.3	NA (9.2)

^a^: binding affinities calculated from binding free energies and averaged over ten independent runs. ^b^ Del Carmine et al., 2004 [88] ^c^ Del Carmine et al., 2002 [75] ^d^: binding affinities calculated from MM-PBSA calculations.

**Table 2 life-12-01393-t002:** Hydrogen bond distances for different β_2_AR-Ligand complexes from AD4 calculations. This analysis was performed on the following ligands: adrenaline (ALE), L-DOPA (DAH) and Droxidopa (DRO). Distances not compatible with hydrogen bonding are shown in bold characters.

Hydrogen Bonds (β_2_AR-Ligand)	Donor-Acceptor Distance (Å)					
	Rigid Model			Flexible Model		
	ALE	DAH	DRO	ALE	DAH	DRO
OD1 (D113)-N (amino)	**3.7** (**4.1**) ^a^	3.0	3.4	2.8	2.8	2.8
OD2 (D113)-N (amino)	2.7 (2.8)	2.5	2.4	2.6	2.5	2.3
OD1 (D113)-O (β-OH)	3.0 (2.8)	- ^c^	2.7	3.1	-	2.9
OG1 (T118)-O (para)	3.3 (**4.4**)	**6.2**	**3.6**	3.4	3.1	3.1
OG1 (T118)-O (meta)	3.2 (**7.0**)	**3.7**	3.3	2.6	2.7	3.1
OG (S203)-O (para)	**4.7** (**3.7**)	**2.8**	**4.3**	2.7	2.6	**4.7**
OG (S203)-O (meta)	**7.1** (3.2)	**7.1**	**5.1**	**5.1**	**4.9**	**7.5**
OG (S204)-O (para)	**6.5** (**5.7**)	3.5	**6.3**	**6.2**	**5.5**	**6.9**
OG (S204)-O (meta)	**8.7** (**4.8**)	**6.2**	**8.7**	**8.0**	**7.8**	**8.8**
OG (S207)-O (para)	2.6 (3.5)	**4.5**	2.9	3.1	2.5	2.7
OG (S207)-O (meta)	**4.0 (6.0)**	2.9	**4.1**	2.8	3.1	**3.9**
OD1 (N312)-N (amino)	2.8 (2.8)	2.9	2.7	3.0	3.0	2.9
ND2 (N312)-O (β-OH)	2.9 (2.8)	-	2.8	**4.9**	-	3.0
ND2 (N312)-O1 (-COO^−^)	- ^b^	2.7	**4.6**	-	2.5	3.5
ND2 (N312)-O2 (-COO^−^)	-	**4.0**	**5.3**	-	**4.0**	**5.0**
OH (Y316)-N (amino)	3.4 (3.5)	3.0	2.9	3.3	2.4	2.4
OH (Y316)-O (β-OH)	**3.9** (3.5)	-	3.5	**4.1**	-	**3.9**

^a^ Hydrogen bonds of adrenaline’s conformation in the β_2_AR X-ray crystal structure (PDB ID: 4LDO) are shown in parentheses. ^b^ COO− group is absent in the adrenaline ligand. ^c^ β-OH group is absent in the L-DOPA ligand.

## Data Availability

Data can be found with A.C.

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
