# Peer review of "L-DOPA and Droxidopa: From Force Field Development to Molecular Docking into Human β2-Adrenergic Receptor"

_life, 2022, doi:10.3390/life12091393_

Round 1
Reviewer 1 Report (Previous Reviewer 3)
Issues have been addressed
Author Response
We thank the reviewer for acknowledging that we have managed to address all the issues present in the first submission of our manuscript.
Reviewer 2 Report (New Reviewer)
The paper shows some interesting behaviors of how adrenaline, L-DOPA and Droxidopa dock with β2AR. I agree with how authors arrived at discussions and conclusions from molecular docking, but I am more concerned about the methodology of molecular docking itself.
1) Figures and pictures should be replaced with vector format images.
2) The units should be consistent. In the FF validation part, kJ is used, while in the rest of results, kcal is used.
3) I am not convinced of how authors avoid randomness from the configuration generation in docking.
4) CM5 is not a typical partial charge deriving method for simulation studies like this. More practically methods, such as ESP, RESP, should be used to study the interactions between ligand and protein.
5) Molecular docking method, either flexible or rigid, is just preliminary or supplementary evidence. More sophisticated methods, such as advanced sampling, alchemical free energy estimation and molecular dynamics simulation, should be used as primary methods, which avoid random errors and consider the temperature effects. Also, molecular docking insufficiently considers the flexibility of the protein chain and functional groups. While docking may show some qualitative observations, the quantitative results are not convincing.
Author Response
The reviewer's comments and suggestions are highlighted in red, the answers are in black, and additions/changes to the original manuscript are in blue.
Reviewer #2
The paper shows some interesting behaviors of how adrenaline, L-DOPA and Droxidopa dock with β2AR. I agree with how authors arrived at discussions and conclusions from molecular docking, but I am more concerned about the methodology of molecular docking itself.
1) Figures and pictures should be replaced with vector format images.
Response:
We thank the reviewer for suggesting the replacement of all the figures and pictures with vector format images. Since all the figures and pictures meet the requirements of the Life journal in terms of format and resolution, the graphical contribution will be kept as it was in the original submission.
2) The units should be consistent. In the FF validation part, kJ is used, while in the rest of results, kcal is used.
Response:
We thank the reviewer for this suggestion. We employed two different units throughout the text because torsional energies are usually reported in kJ/mol in FF development articles, while the binding free energies results, which were obtained using Molecular Docking calculations and Molecular Dynamics (MD) simulations, are usually reported in kcal/mol in the literature. To avoid this inconsistency, we have added the following line with the conversion factor in the captions to the figures of the main text of the manuscript:
Energies are reported in kilojoules per mole (1 kJ/mol = 0.24 kcal/mol).
3) I am not convinced of how authors avoid randomness from the configuration generation in docking.
Response:
We appreciate the reviewer's feedback. Ten separate molecular docking runs have eliminated unpredictability in the formation of configurations. For both rigid and flexible β2AR models, 400 poses were created by employing a maximum generation count of 27,000 and an energy evaluation of 5x107, as previously described for docking dopamine to the D2DR receptor. The resulting docked conformations were subjected to a Virtual Screening analysis utilizing the AD4 pythonsh function and a clustering tolerance of 2 Å root mean square deviation (RMSD).
4) CM5 is not a typical partial charge deriving method for simulation studies like this. More practically methods, such as ESP, RESP, should be used to study the interactions between ligand and protein.
Response:
We thank the reviewer for this suggestion. The CM5 partial charge deriving method was used for the development of FF parameters of each ligand. CM5 charges are increasingly popular in the development of force fields because they are available in several electronic structure codes, are nearly invariant for different quantum chemical models (contrary to ESP, etc.) and reproduce very well the molecule dipole moment. The usage of CM5 partial charges is also supported by the following reference:
Sami, S., Menger, M. F. S. J., Faraji, S., Broer, R., & Havenith, R. W. A. (2021). Q-Force: Quantum Mechanically Augmented Molecular Force Fields. Journal of Chemical Theory and Computation, 17(8), 4946-4960. https://doi.org/10.1021/acs.jctc.1c00195
5) Molecular docking method, either flexible or rigid, is just preliminary or supplementary evidence. More sophisticated methods, such as advanced sampling, alchemical free energy estimation and molecular dynamics simulation, should be used as primary methods, which avoid random errors and consider the temperature effects. Also, molecular docking insufficiently considers the flexibility of the protein chain and functional groups. While docking may show some qualitative observations, the quantitative results are not convincing.
Response:
We totally agree with the reviewer on the limitations of molecular docking calculations and the possibility of employing some of the suggested sophisticated methods. We have also performed microsecond long MD simulations of each ligand bound to the receptor. Moreover, we have also calculated the binding free energy of each ligand using MM-PBSA calculations over the equilibrated trajectories of MD simulations. Both approaches have provided a more detailed picture of how the flexibility of the whole protein and, especially, the residues of the binding pocket of human β2AR plays an important role in the receptor interaction with each ligand.

Round 2
Reviewer 2 Report (New Reviewer)
Please make proper changes in the MS to reflect the addressment of my reviews.
Author Response
On response to the reviewer’s comments of our latest submission, with the manuscript number LIFE-1817394. The reviewer’s comments and suggestions are highlighted in red, the answers are in black, and additions/changes to the original manuscript are in blue.
Reviewer #2
Please make proper changes in the MS to reflect the addressment of my reviews.
Response:
1) We highly appreciate the suggestion, according to our responses to round 1 comments, we prefer to retain the original format of the Figures.
2) The conversion factor between kJ/mol and kcal/mol has been added in the captions of Figures 3-8.
3) The answer to this point has been reported in the Molecular Docking protocol of the Materials and Methods section of the revised manuscript.
4) The rationale behind the choice of CM5 charges has been added on page 3 of the revised manuscript.
5) The modifications of the text and the additional computations performed were already described in the answers to round 1 comments.
This manuscript is a resubmission of an earlier submission. The following is a list of the peer review reports and author responses from that submission.
Round 1
Reviewer 1 Report
Manuscript ID: life-1533695
Title: L-DOPA AND DROXIDOPA: FROM FORCE FIELD DEVELOPMENT TO MOLECULAR DOCKING INTO HUMAN β2-ADRENERGIC RECEPTOR
The manuscript authored by Andrea Catte *, Akash Deep Biswas, Giordano Mancini, Vincenzo Barone. Showed the derivation of L-DOPA and Droxidopa OPLS all atom (AA) force field (FF) parameters via quantum mechanical (QM) calculations, molecular dynamics (MD) simulations in aqueous solutions of the two catecholamines and the molecular docking of both ligands into rigid and flexible β2AR models.
However, a significant question arose after reading the manuscript:
- In summary, it is mentioned that a molecular dynamics of 1 microsecond was carried out, but in the materials and methods section, only 100 ns are stipulated. What is the correct data?
- In line 142, the construction of a model by homology of the β2AR receptor using 2RH1 as template is mentioned; In the results and discussion section the validation of the same is not shown, a fact that is important since all the work is based on whether the model obtained is reliable.
- The authors should include the parameterization of adrenaline and compare it with that obtained for L-DOPA and Droxicpa.
- The authors can discuss why different conformations of adrenaline were obtained from both flexible and rigid redocks; giving biological relevance to this discussion.
- They can include in Table 1 the calculation of the binding affinity (BFEs) from the molecular dynamics studies for all the ligands and thus compare it with the docking and experimental results.
- The authors should discuss which of the methodologies used in their work is better and why (flexible, rigid docking with AD4 or vina and DM).
- The authors should include the minimum analyzes of the DM studies of the complexes β2AR-adrenaline, β2AR-L-DOPA, and β2AR-Droxidepa (RMSD vs time, Residues that interaction as a function of time, binding energy calculations obtained from MM- PBSA)
Additionally, I am attaching a pdf file to clarify some grammar observations

Reviewer 2 Report
The current study is an extension of Biswas et al, the current manuscript is well-designed. The docking and simulation data support the crystallographic and in vitro experimental data.
The motivation for the current study is not highlighted clearly, discussion can also use the relevance of this finding for a broader audience.
Reviewer 3 Report
In this work, authors provided the binding mechanisms at the molecular level between the adrenergic receptor protein (AR) and two drug candidates comparing with the native adrenaline substrate. Authors started with performing an accurate force field parameterization before performing extensive series of MD simulations and compare the results of the newly developed parameters with the standard CHARMM forcefield.
In my opinion, the quality of this manuscript was onto the reliability of computational methods where the ample amount of trajectories are provided to strengthen the assumption. However, more analysis, and discussions, along with some changes might be needed to validate the results and enhance the significance of this paper.
- RMSD calculations for both protein and ligand are needed to ensure the stability of global protein structures and monitor the difference of binding postures between docking and MD.
- Apart from molecular docking, binding affinity from MD trajectories should also be quantified by the MM/PBSA calculation. The per-residue decomposition of MM/PBSA energy should also confirm the hydrogen bonding network analysis.
- In my opinion, it is alright to mention about the forcefield development but it should not be the highlight of this paper, as the molecular mechanisms of the two potential inhibitors should be more significant for people in pharmacology field. I suggested that the authors could put some parts of the forcefield development into the supplementary data and be more focused on how some molecular features of the two inhibitors improved the protein binding in comparison with the native substrate (quantitative data from per-residue MM/PBSA should also help).
Reviewer 4 Report
Authors have recently published similar results in Biophysical Journal (https://doi.org/10.1016/j.bpj.2021.11.007 ) This reviewer doesn't see any improvement over the previously published work except the development of the force field. Hence I cannot recommend the manuscript in its current form.